# Dynamics of Ultrafast Phase Transitions in (001) Si on the Shock-Wave Front

**DOI:** 10.3390/ijms23042115

**Published:** 2022-02-14

**Authors:** Evgenii Igorevich Mareev, Fedor Viktorovich Potemkin

**Affiliations:** 1Faculty of Physics, M. V. Lomonosov Moscow State University, Leninskie Gory bld.1/2, 119991 Moscow, Russia; potemkin@physics.msu.ru; 2Institute of Photon Technologies, Federal Scientific Research Centre “Crystallography and Photonics”, Russian Academy of Sciences, Pionerskaya 2, Troitsk, 108840 Moscow, Russia

**Keywords:** phase transitions, molecular dynamics, shock waves, X-ray diffraction

## Abstract

We demonstrate an ultrafast (<0.1 ps) reversible phase transition in silicon (Si) under ultrafast pressure loading using molecular dynamics. Si changes its structure from cubic diamond to β-Sn on the shock-wave front. The phase transition occurs when the shock-wave pressure exceeds 11 GPa. Atomic volume, centrosymmetry, and the X-ray-diffraction spectrum were revealed as effective indicators of phase-transition dynamics. The latter, being registered in actual experimental conditions, constitutes a breakthrough in the path towards simple X-ray optical cross-correlation and pump-probe experiments.

## 1. Introduction

Silicon is the most studied semiconductor; however, its high-pressure phases due to new theoretical and experimental techniques have only been actively examined in recent years [1,2,3,4]. Nowadays, it is possible to accurately investigate the high-pressure behavior of materials, such as structural properties, solid-to-solid phase transitions, and dynamical properties. Si under static pressure transforms from the diamond (Si-I) to the metallic β-Sn (Si-II) structure at around 11 GPa. Under higher pressure, the hexagonal close-packed (hcp-Si-VII), face-centered cubic (fcc-Si-X) and other structures have been examined as possible high-pressure phases of Si [2,5,6,7,8]. To summarize, silicon has eleven (or even more) distinct stable and metastable crystalline phases at high pressures. Numerical calculations have predicted the structural phase transition from cubic diamond to β-Sn (~11 GPa) and from β-Sn to hcp for Si (about 40 GPa) [3,5,9,10,11]. The hcp phase of Si is stable at about 40 GPa in diamond-cell experiments [2]. However, despite being reasonably well-understood under hydrostatic conditions, under dynamic loading, the phase transitions in Si remain a subject of debate [12]. Moreover, depending on the amplitude and time duration of loading, the final phase of Si is varied from amorphous to *Ibam*, BC8, hex. diamond (Si-IX), etc. [13,14]. The presumptive tree of phase transitions in Si primarily depends on the velocity of pressure release and do not take into account the role of temperature that would probably arise in the experiments, both in the case of laser ablation (when the laser pulse is directly focused onto the sample’s surface [15,16]) or shock-wave impact. Under such conditions, the phase transitions may be masked by the melting of Si and could not be detected post-mortem. Nowadays the time-resolved method that is applicable to the direct detection of phase transitions is limited by the time-resolved Raman scattering and X-ray diagnostics [17,18]. Recent progress in the generation of X-rays, both from free-electron lasers (FEL) and laser plasma, or from the high harmonic generation, which allows for the retrieval of the ultrafast dynamics of phase transitions [19,20,21]. Using such sources, the time-resolved experiments were successfully applied to the study of lattice dynamics [22,23], phase transitions [24,25], and phonon dynamics [26]. However, the experiments did not provide direct information about structural dynamics and atom motion. To “build a bridge” between observations in the experimental macroscopic parameters (such as Raman or XRD spectra) and the microstructure of the investigated sample, a numerical simulation is commonly applied [27]. Therefore, it is extremely important to perform numerical simulations to reveal the lattice dynamics that occurred between the start of the process and the formation of the final phase.

Under static conditions, the phase transitions in Si are widely investigated using the ab initio total-energy-pseudopotential method [5]; however, this method is not applicable to dynamical phase conditions. Recently, molecular dynamics were used to investigate the phase transitions induced by the shock wave in Si [11]. The authors demonstrated that the shear stress is relieved by an inelastic response associated with a partial transformation to a new high-pressure phase, where both the new phase and the original cubic-diamond phase are under close hydrostatic conditions. However, the dynamics of the phase transition were not investigated. In experiments, the multi-megabar pressures are usually achieved by high-intensity, ultrashort laser pulses [28,29,30]. For example, by combining a free-electron-laser (FEL)-based X-ray-diffraction geometry with laser-driven compression, the lowering of the hydrostatic phase boundary in elemental silicon was demonstrated [4]. The authors demonstrated melting above 14 GPa, which was previously ascribed to a solid-to-solid phase transition. For example, it is possible to achieve such pressures in laser shock peening experiments [31]. In the framework of the approach, the shock wave is generated in a buffer media (for example, water), and this shock wave affects the Si sample. This leads to the increase in the interaction region, which simplifies the X-ray-diffraction (XRD) experiments [31]. 

In the current work using molecular dynamics (MD), we investigated solid-to-solid phase transitions in Si under the shock-wave impact. We simulated both static and dynamic cases. A static case was used as a reference for determining a phase state under dynamic loading and verifying the applied numerical model. We successfully modeled phase transitions to the β-Sn and hcp phases under 11 and 40 GPa, respectively. We observed the XRD spectrum, atomic volume, and centrosymmetry data that were used to identify new phases in the dynamic conditions. Then, we demonstrated that an ultrafast (<0.1 ps) phase transition to β-Sn occurs on the shock-wave front. The phase transition was recorded as the “jump” in the atomic-volume, XRD-spectrum, and centrosymmetric-parameter (CSP) changes. A detailed description of the used terms is given in the Section 3.

The critical subject of the current manuscript is an investigation of the non-stationary extreme state of matter with a short lifetime in solids using MD. This approach is prevalent for calculating non-stationary processes because it allows for dynamically changing the external impact on the system (for example, by adding shock or laser impact) [32]. Molecular-dynamics simulations aim to obtain a deep understanding of molecular interactions and trajectories that could not be understood otherwise [33]. MD allows us to directly determine the effect of the microscopic properties on the macroscopic parameters of the system. 

The complete details about the applied method can be found in the computation details section. In the first set of simulations, we investigate the phase transitions under static pressure loading. This set of simulations is used as a reference to obtain well-known high-pressure phase transitions in Si, and to retrieve the XRD spectrum and compare it with the experimental spectrum, for example, of the synchrotron facility [34]. We modeled the uniaxial shock-wave propagation along the (001) direction in single-crystal silicon in the following simulations. 

## 2. Results and Discussion

### 2.1. Stationary Conditions

The first set of simulations was performed under hydrostatic conditions. The simulations were performed with a 5 GPa step far from the phase-transition pressure and a 0.5 GPa step in the vicinity of the presumptive transition pressure. These simulations aimed to retrieve the atomic volume, XRD spectrum, and centrosymmetric parameter for different phases of Si. We determined that Si is transformed from cubic diamond (atomic volume: 20 A^3^) to the β-Sn phase under the pressure of about 11 GPa. The phase is stable until reaching 45 GPa. At this pressure, the structure of Si is transformed to hcp (see Figure 1). The obtained data is in good agreement with the literature data [2]. The cross-section along the z-direction demonstrates that the structure of Si changes (shown by lines in Figure 1b). The non-symmetric nature of the structure change is raised because the phase transition starts from the inhomogeneities in the structure (no phase transitions were observed in the simulations of the ideal Si crystal). Therefore, the directions of the crystallites are random. This leads to a decrease in the peaks’ intensities in the XRD spectrum.

The phase transition in Si leads to the jump in the atomic volume, centrosymmetry parameter, and XRD spectrum. In the other cases, smooth changes in these parameters occur. For example, under normal conditions, the atomic volume of Si is about 20 A^3^, as shown in the histogram presented in Figure 2a. The phase transition occurred under pressure of about 11 ± 1 GPa, leading to the jump in atomic volume (~16 A^3^) as shown in Figure 2a. The phase transition leads to the change in the centrosymmetric parameter: new peaks in the histogram appear (see Figure 2b). Moreover, the XRD spectrum is significantly changed: the amplitude of all peaks is decreased, and new peaks (for example, at 35°) appear. The following pressure increase does not lead to the jump in the atomic volume or centrosymmetry parameter (until reaching 45 GPa). The mean value of the atomic volume is smoothly shifted toward lower values. The histogram of the centrosymmetry parameter transforms from a one-peak structure in the cubic-diamond phase to the three-peak structure in the β-Sn phase. The most contrasting picture of a three-peak structure is observed under 30 GPa pressure. The β-Sn phase is stable under pressures below 45 GPa; the atomic volume in this phase can be as low as 14 A^3^. Therefore, the change in Si structure leads to the jump in atomic volume, as well as the changes in the centrosymmetry parameter and XRD spectrum. The latter is the most important because it can be observed experimentally.

### 2.2. Non-Stationary Conditions

The next series of simulations were performed in order to retrieve the ultrafast dynamics of the phase transition induced by the shock impact, which is demonstrated in Figure 3. From the previous simulation, we observed that the transition to the β-Sn phase is accompanied by a jump in the atomic volume to 16 A^3^ (see Figure 3a,b), a change in centrosymmetry parameter, and new peaks in the XRD spectrum. First, we obtained the profile of the particle-velocity (the velocity of Particle A in a medium as it transmits a wave) z-component and atomic volume for the piston velocities of 3 km/s and 1 km/s (Figure 3c,d). Under the impact of a 3 km/s piston, the profiles demonstrate a jump in the atomic volume up to 14.5 A^3^. The volume reaches this level on the shock-wave front. After passing the shock-wave front, the volume relaxes to 20 A^3^. Thereby, a running along the z-axis region of the β-Sn phase arises (see Figure 3). The amount of new phase growth during shock-wave propagation is due to the broadening of the shock-wave front. Under 1 km/s piston speed, the amplitude of the shock wave is not enough to achieve β-Sn phase (there is no jump to 16 A^3^ atomic volume).

During propagation, the amplitude of the shock wave drops and the width of the shock-wave front is increased. The peak shock-wave velocity is about 15 km/s, which is similar to the results obtained for (111) Si [35]. The shock-wave velocity rapidly (about 2–3 ps) decays and the shock wave loses about 75% of its energy, and the corresponding particle velocity decays similarly (see Figure 4). The phase transition mainly occurs during this time interval (when the shock wave loses 75% energy). The dependence between shock and particle velocity is presented in Figure 4 (see inset) and represents its shock adiabat [35]. 

Under pressures achieved under such an impact, a compression of Si becomes inelastic and exhibits the “anomalous” elastic waves [36]. The additional increase in piston velocity leads to the destruction of the first several layers of the sample. The appearance of these waves could be a result of ultrafast phase transitions to the β-Sn phase, which exists only on the shock-wave front. After passing about 30μm, the amplitude on the shock-wave front is not enough to induce a transition to the β-Sn phase; therefore, the phase transition occurs only in the vicinity of the sample boundary. Under the impact of a 1 km/s shock wave, the shock-wave pressure is not enough to induce the transition of the detectable volume of Si (~10–20 atoms are in β-Sn phase). Such a small amount of material in a new phase demonstrates the extremely small probability of transition to the β-Sn phase (under 1 km/s particle velocity) and to our knowledge, there are no experimental techniques able to detect this new phase. Therefore, it could be said that the phase transition to the β-Sn phase could not occur under such conditions. 

Moreover, we retrieved the histograms of the atomic-volume distributions, the distribution of the centrosymmetry parameter, and the XRD spectrum of the volume affected by the shock wave (30 × 30 × 50-unit cells i.e., a layer containing 50 atom layers). The evolution of these distributions is presented in Figure 5. It demonstrates that the shock wave perturbates the initially Gaussian distribution of the atomic volume. The propagation of the shock wave (see Figure 5) is accompanied by the appearance of a second peak in the vicinity of 16 A^3^. The peak at 20 A^3^ rapidly transforms to 16 A^3^. It indicates the phase transition to the β-Sn phase, and the first peak shows that the cubic-diamond structure of Si vanishes. A beginning of the transformation to the beta-Sn phase after 200 fs is observed. The change in the centrosymmetry parameter (CSP), atomic volume, and XRD spectrum associated with the transition to the β-Sn phase is obtained during propagation of the shock wave (see Figure 5). This leads to the appearance of new XRD components in the XRD spectrum at 31° and 47° (see Figure 5e,f). In contrast to the atomic volume and centrosymmetry parameter, the XRD spectrum can be detected experimentally. After the shock wave passes, the structure relaxes to its initial phase, which continues at about ~1 ps because the β -Sn phase is not stable at ambient conditions. The relaxation of the system is slower because the shock wave abruptly disorders the atoms’ positions, which leads to the appearance of the new phase. Then, when the shock pressure drops, the atoms begin to drift to their initial state. Because the displacement is comparatively large (about one-half of interatomic distance) the volume distribution (see Figure 5a) becomes broader; the atom could not return to its exact initial position. If the amplitude of the shock wave is higher, then the shifts of the atoms from their initial positions will be so large that the structure will be broken, the lattice could be destroyed, and the Si will become amorphous. The amplitude of the peak in the atomic-volume distribution after shock-wave impact is lower and its width is higher than in the initial state. It is caused by higher atom-oscillation amplitudes due to higher kinetic energy accumulated during the passage of the shock wave. The phase transition also leads to the change in the XRD spectrum that can serve as an indicator in the experiments. Compared with the stationary case, the difference in the centrosymmetry parameter is not contrasting enough (see Figure 5c,d). Under 1 km/s piston velocity, no significant jumps in the atomic volume were detected. The analysis shows that only a small amount (less than 10^−6^) of atoms are in the β-Sn phase, and in the histograms it manifests itself as a broadening a peak at 20 A^3^ to lower volumes (see Figure 3a), in contrast to 3 km/s piston velocity, for which a new peak at 16 A^3^ is formed. Such a small amount of the new phase in our opinion is not enough to assert the possibility of a phase transition of α-diamond => β-Sn under 1 km/s piston velocity. 

As the last part of the analysis, we obtained the intensity (z-t) maps of the shock-wave velocity, shock-wave pressure, and atomic volume (see Figure 6) in order to better localize the phase transition in the time-and-space domain. The intensity maps show that a phase transition could occur only in small regions in the z-t diagram. The maximal achieved pressure is 27.4 GPa for 3 km/s piston velocity and 8.1 GPa for 1 km/s. The areas marked as red and orange in Figure 6c,d indicate the transition to the new phase with an atomic volume of about 10 A^3^. The pressure necessary for the transition is about 11 GPa, which is in good agreement with the threshold pressure obtained under stationary conditions. As can be seen from Figure 6, the area of the phase transition is increased with time as a result of shock-wave-front broadening. However, when the pressure drops below the threshold value, the phase transition will not occur. It is important to note that an increase in initial perturbation (from δ-function to a Heaviside or exponential-decay function) does not significantly change the observed phenomena. It will increase the phase-transition region due to the increase in the shock-wave front.

MD shows that the further growth of the piston velocity (above 3 km/s) will lead to the destruction of the Si lattice, and the structure in the thin layer will not relax to the initial state. The relaxation velocity depends on the initial amplitude of the shock wave and the width of the shock-wave front. An increase of the piston velocity above 3 km/s (or the width of the shock-wave front) leads to the destruction of the Si lattice and amorphous Si is formed (at least at the several microseconds time interval). The amorphous region for 4 km/s piston velocity is about 20 A deep. This region is presented in Figure 7c as a growing laser with a temperature (a measure of the kinetic energy of molecules) ~ melting temperature. On a microsecond timescale and under such conditions, the phase transition becomes irreversible, and it can be detected post-mortem.

The possible source of the lattice destruction could be its overheating (temperature above the melting temperature of 1683 K). The temperature of the shock front grows due to the adiabatic compression of the material. Both in the cases of 3 km/s and 4 km/s of piston velocity, the temperature on the shock front overcomes the melting threshold (see Figure 7). Nevertheless, the instantaneous temperature itself does not lead to the melting of the material because the process of the lattice destruction could not occur on the sub-ps timescale while the temperature of the lattice has to be higher than the melting point of the material. It takes place when the piston velocity is 4 km/s. In this case, under the shock impact, the temperature of several boundary layers is higher during all simulation periods (see Figure 7c), which naturally leads to the amorphization of the Si, which can be detected post-mortem. 

## 3. Methods

In this work, we used the LAMMPS computation package [27] to perform two sets of MD simulations. In the first set of simulations, we investigated the phase transitions under static-pressure loading. This set of simulations was used as a reference to obtain well-known high-pressure phase transitions of Si and to retrieve the XRD spectrum and compare it with the experimental one. The simulation was performed on a 30 × 30 × 30 unit cell with about 225,000 atoms. Periodic boundary conditions were imposed on three dimensions.

Under periodic boundary conditions and in an ideal cell, it is practically impossible to reproduce phase transitions due to the absence of the new-phase seeds. In experiments, similar phenomena could be observed in oversaturated solutions or overheated matter. Therefore, to simulate phase transitions in MD several “tricks” are used. For this purpose, two regions with different phases are created and external parameters are varied until one phase remained. Another approach is to use triclinic boxes or manually create inhomogeneities. Therefore, we added about 100 randomly distributed vacancies to the cell. The number of vacancies weakly affects the result and only changes the simulation time, and the phase transition could be reproduced even with one vacancy. In our simulation, we increased the number of vacancies in such a way that a phase transition occurred in half the simulation time. These vacancies become the center of the phase transition under high-pressure loading. The number of vacancies is lower than 0.05% of the sum number of atoms, and their role is only to become the seeds for a new phase. Before the impact of the external pressure, we equilibrated the system at 305 K for 10 ps (npt-ensemble). Then we added the static pressure to each boundary of the system using the npt-ensemble with the Rahman–Parinello method [37] and a triclinic box. Periodic boundary conditions were imposed on the two dimensions transverse to the shock-propagation direction. The interatomic interactions were the Tersoff-like potential [38], which was parametrized by [9] to be used in the simulations. This potential was previously and successfully used to simulate various Si properties in stationary [39,40] and non-stationary conditions [5,7,11]. The structure of the system was visualized by the Ovito software package [41]. The simulations were performed with a 5 GPa step far from the phase-transition pressure and a 0.5 GPa step in the vicinity of the presumptive transition pressure.

We calculated the atomistic volume using Radical Voronoi tessellation using the Voro++ library [42]. For each pressure, we also calculated the centrosymmetric parameter and the XRD spectrum [43]. The centrosymmetry parameter (CSP) quantifies the local loss of centrosymmetry at an atomic site. The CSP of an atom having N nearest neighbors is defined as:(1)$$CSP=\sum_{i=1}^{N/2}{\left|{\vec{r}}_{i}+{\vec{r}}_{i+N/2}\right|}^{2},$$

Each symmetry group (for example, cubic diamond) has a unique distribution of *CSPs* (for an ideal cell *CSP* = 7 with a basis of 2) where ${\vec{r}}_{i}$ and ${\vec{r}}_{i+N/2}$ are vectors from the central atom to a pair of opposite neighbors; the CSP indicates the number of atomic neighbors along a specific basis. Due to the Gaussian distribution of atomic position in the actual crystal, the CSP also has a Gaussian distribution. Affine deformation of the lattice does not change its degree of centrosymmetry at all. Each phase transition changes the centrosymmetry parameter. Therefore, it can be applied as an indicator of the phase transition.

The Voronoi decomposition [44] serves as a simple geometric method to determine the near neighbors of an atom by considering the faces of the Voronoi polyhedron enclosing the particle. The geometric shape of the Voronoi polyhedron reflects the characteristic arrangement of close neighbors [41,45]. The result of the analysis can give information about the atomic volume and the lattice type [41]. The X-ray-diffraction intensity (XRD spectrum) was calculated as described in [27] in a mesh of reciprocal lattice nodes defined by the entire simulation domain using simulated radiation of wavelength lambda.
(2)$$I=Lp(\theta )\frac{F*F}{N},$$
(3)$$F(k)=\sum_{j=1}^{N}f_{j}(\theta )\exp(2\pi i\vec{k}\cdot {\vec{r}}_{j}),$$
(4)$$L_{p}\left(\theta \right)=\frac{1+\cos^{2}(2\theta )}{\cos(\theta )\sin^{2}(\theta )}$$
(5)$$\frac{\sin(\theta )}{\lambda }=\frac{\left|\vec{k}\right|}{2}$$

Here, ***k*** is the reciprocal lattice node’s location, ${\vec{r}}_{j}$ is each atom’s position, ***f***_j_ are atomic scattering factors, *L_p_* is the Lorentz-polarization factor, and *θ* is the scattering angle of diffraction. 

The subsequent simulations modeled the uniaxial shock-wave propagation along the (001) direction in single-crystal silicon. The shock waves with propagation velocity U_s_ were launched into the system by the impact of a piston only a few unit cells wide and with constant atomic velocity U_p_ as a delta-function of time, after the thermalization of the sample at 305 K for 10 ps. Setting the particle velocities corresponding to the several first atomic layers corresponds to the boundary conditions during shock-wave propagation when the particle velocity in one media equals the particle velocities in another. Periodic boundary conditions were imposed on the two dimensions transverse to the shock-propagation direction. The sample cross-section was 30 × 30 unit cells and 390 unit cells along the shock-wave propagation direction (the z-axis), corresponding to 2,809,800 atoms. For all values of U_p_, the shock front does not reach the rear end of the sample. Piston velocities were varied between U_p_ = 1 km/s and U_p_ = 3 km/s. The samples were considered perfect defect-free crystals because the shock wave destroys the symmetry of the lattice and acts itself as the center of the phase transition. For the XRD simulation we changed the box from triclinic to orthorhombic.

## 4. Conclusions

We retrieved the dynamics of sub-ps phase transition in Si (from cubic diamond to β-Sn phase) induced by the shock wave using molecular dynamics. The ultrafast phase transition occurs only on the shock-wave front if pressure overcomes 11 GPa at the sub-ps timescale. The phase transition is accompanied by the jump in the atomic-volume (from 20 A^3^ to 16 A^3^), centrosymmetry-parameter, and the XRD-spectrum changes (appearing of new peaks at 31° and 45°), which can be detected in the experiments. The new phase appears during shock-front propagation and vanishes after unloading; therefore, it can be seen in the optical pump, which is an X-ray-probe experiment under nanosecond and femtosecond impact. The lattice heating induced by the shock-wave propagation under high piston velocities (~4 km/s) leads to the destruction of the lattice and could be detected post-mortem. The proposed concept, being registered in actual experimental conditions, constitutes a breakthrough in the path towards simple X-ray optical cross-correlation and pump-probe experiments.

## Figures and Tables

**Figure 1 ijms-23-02115-f001:**
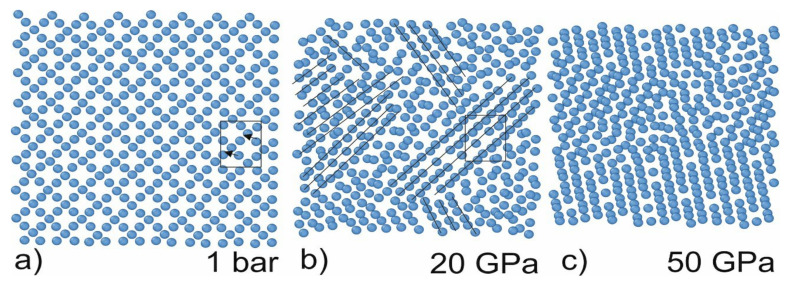
Cross-sections of the simulation cell along the z-axis under different pressures. The height of the layer is 4A (two atomic layers). The arrows indicate atom shifts during phase transition: under hydrostatic pressure, central atom shifts (arrows show the shift direction) and form a new lattice. The new phases arise in the vicinity of the defect; therefore, the direction of the lattice is random. The black lines indicate the rows in the same direction. (**a**) α-diamond phase (**b**) β-Sn phase (**c**) hcp phase.

**Figure 2 ijms-23-02115-f002:**
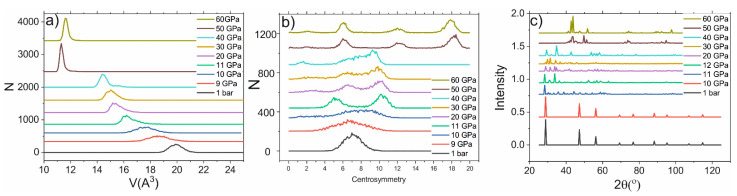
(**a**) Histogram of atomic-volume distribution under different pressures. (**b**) Histogram of centrosymmetry-parameter distribution under different pressures (**c**) XRD spectra under different pressures.

**Figure 3 ijms-23-02115-f003:**
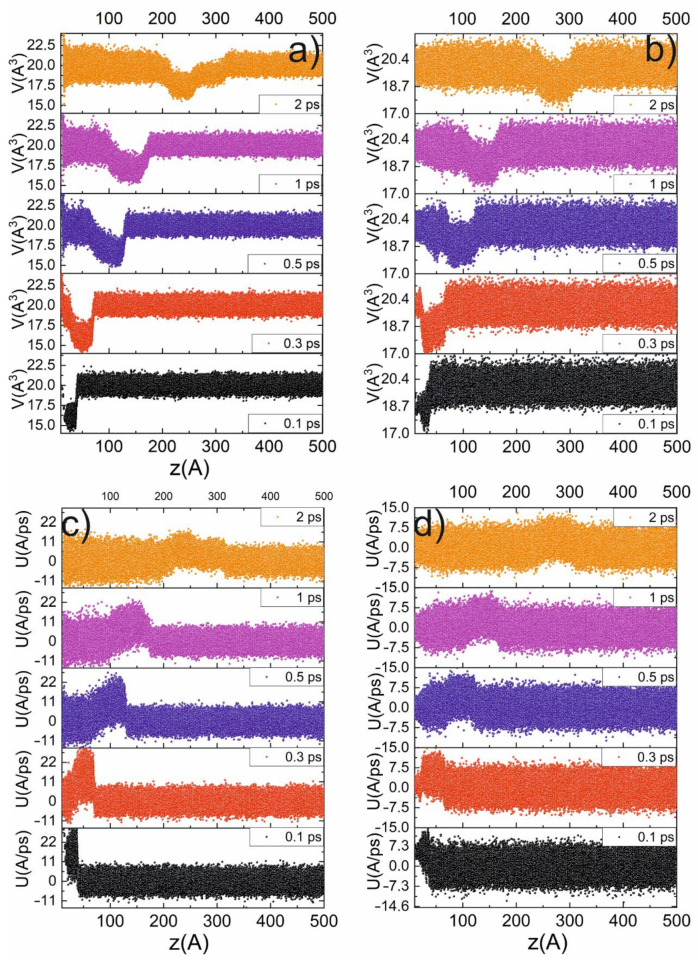
Profiles (along z-axis) of atomic volume (**a**,**b**) and particle velocity (**c**,**d**) at different time delays for 3 km/s (**a**,**c**) and 1 km/s (**b**,**d**) piston velocities.

**Figure 4 ijms-23-02115-f004:**
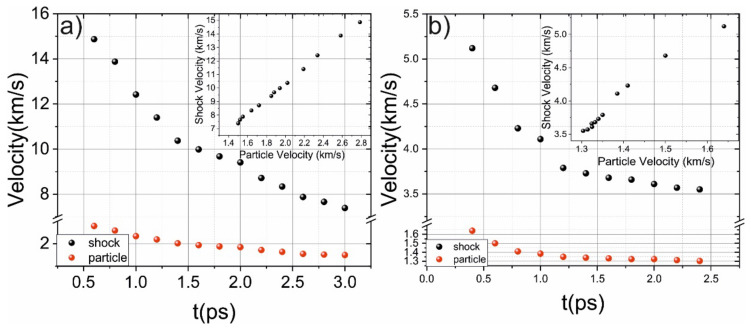
(**a**) Evolution of shock (black dots) and particle velocity for 3 km/s piston velocity (**a**) and 1 km/s piston velocity (**b**). The insets demonstrate dependence between shock and particle velocity.

**Figure 5 ijms-23-02115-f005:**
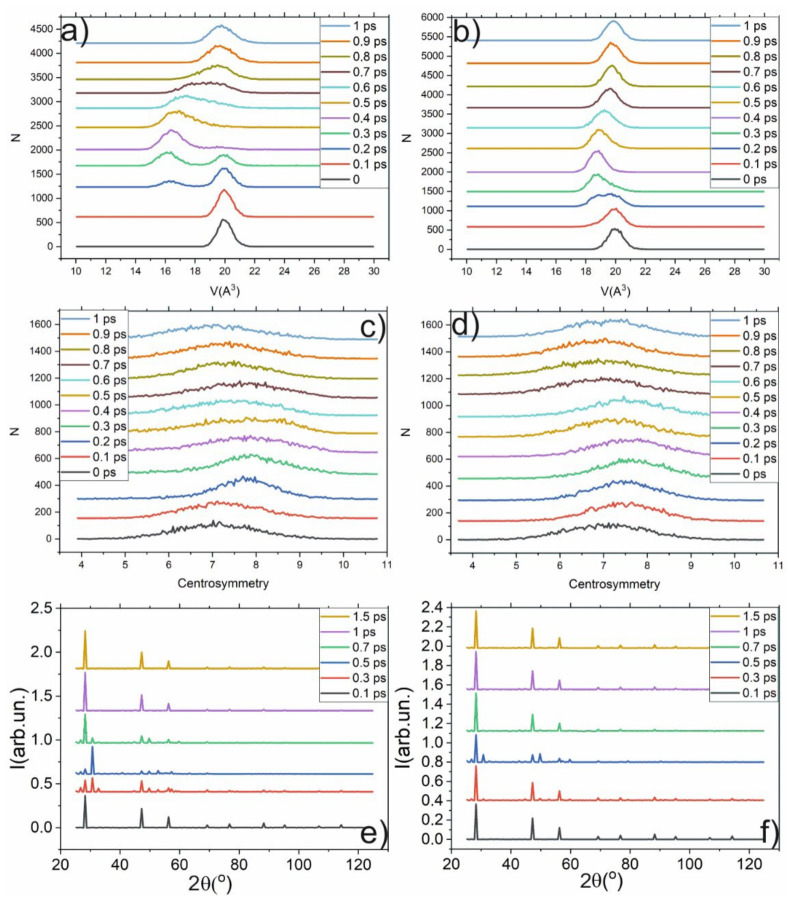
Evolution of the atomic volume distribution (**a**,**b**), distribution of CSP (**c**,**d**) and XRD spectrum in the case of 3 km/s piston velocity (**a**,**c**,**e**) and 1 km/s piston velocity (**b**,**d**,**f**) in the volume affected by shock wave (30 × 30 × 50 unit cells).

**Figure 6 ijms-23-02115-f006:**
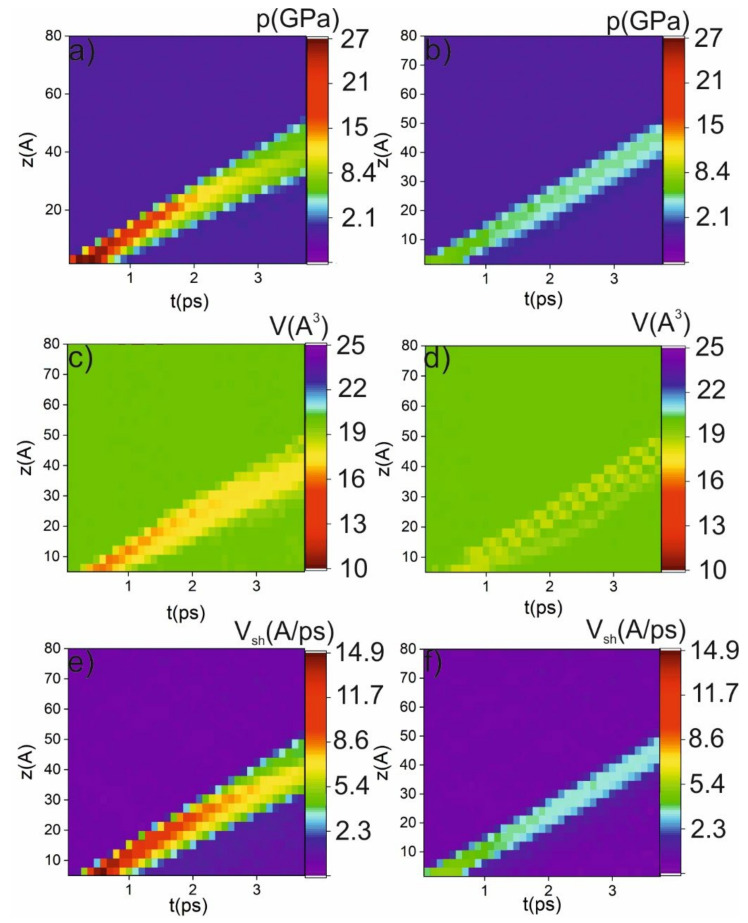
Intensity maps of pressure (**a**,**b**), atomic volume (**c**,**d**) and shock-wave velocity (**e**,**f**) in the case of 3 km/s piston velocity (**a**,**c**,**e**) and 1 km/s piston velocity (**b**,**d**,**f**).

**Figure 7 ijms-23-02115-f007:**
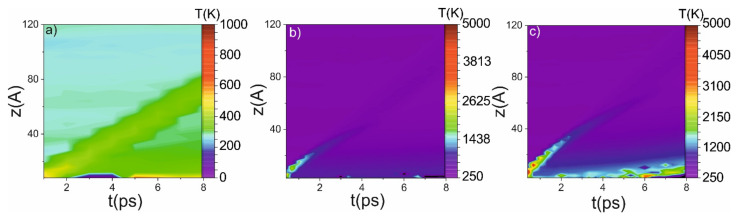
Intensity maps of temperature in the case of 1 km/s (**a**), 3 km/s (**b**) and 4 km/s (**c**) piston velocity. The initial temperature is 305 K, the melting temperature is 1683 K.

## Data Availability

Data underlying the results presented in this paper are not publicly available at this time but may be obtained from the authors upon reasonable request.

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
