# Peer review of "Dynamics of Ultrafast Phase Transitions in (001) Si on the Shock-Wave Front"

_ijms, 2022, doi:10.3390/ijms23042115_

Round 1
Reviewer 1 Report
I attached a pdf file for my comments and suggestions for authors.

Author Response
First of all, we want to thank the Reviewer for their work. All changes in the manuscript are presented in the red-line version of the manuscript. The answers to the issues are given below.
Point 1: Lines 54-62: The authors explained time-resolved XRD with shock-wave compression generated by ultrashort intense lasers. This kind of experiments (e.g. metallic films, nanocrystals) were already demonstrated at the XFEL facilities [A. M. Lindenberg, Annual Review of Materials Research, 47, 428 (2017)]. A brief description of the resent experimental studies about high-pressure/shock-induced transformations may provide the feasibility of the time-resolved XRD for ultrafast transitions of silicon crystals.
Answer: We expanded the introduction and provide additional references on the time-resolved experiments, including FEL facilities.
Point 2: Line 110: Fig. 2b -> Fig. 2a
Answer: We fixed this text fragment.
Point 3: Line 111 and figure 2: The authors state the atomic volume is jumped down under the specific pressure ~11+/-0.5 GPa. I recommend to show 9 (or 10) GPa results on figure 2 in order to emphasize the jumpdown phenomena.
Answer: We added additional graphs to the figure to illustrate the phenomena.
Point 4: Line 134: the definition of the particle velocity is unclear.
Answer: We add the definition of particle velocity to the text: “(the velocity of a particle in a medium as it transmits a wave)”
Point 5: The scatter plots are very indistinct. I recommend to shift each profile like figures 2 and 5.
Answer: We redrawn Figure 3.
Point 6: Line 159: I don’t understand the meaning of “detectable volume of Si” in this line. Q1) At 3 km/s piston speed, the Si crystals have been transformed until ~30 μm depth. At 1 km/s piston speed how deep the shock wave front can keep the enough amplitude to cause the transitions? Q2) Figures 5b and 5f seem to be not shown any transformation to β-Sn. And also in line 197, the authors described that the maximal achieved pressure is 8.1 GPa for 1 km/s. I suspected that the 1 km/s piston velocity cannot causes β-Sn transformation on any region of Si. Q3) At figure 6, the authors clime the recrystallization which means the β-Sn phase relax to amorphous one. Does the detectable volume of Si mean the thickness of the region keeping the β-Sn phase during the shock wave propagation? If so, the authors have to describe the detectable volume of Si as the thickness of β-Sn phase based on the figure 6. I suppose the evaluation of detectable volume is extremely important for feasibility of this kind of experiment.
Answer: By detectable volume, we mean volume that could be detected in the experiments. We added the description of the term to the text of the manuscript. We also want to mention. That de jure, shock wave with piston velocity 1km/s causes phase transition, however, de facto the probability of such transition is extremely low, that no one of known experimental techniques could detect the new phase. Thereby following the reviewer’s comment we added the following text fragment to the text: “….. Si (~10-20 atoms are in b-Sn phase). Such a small amount of material in a new phase demonstrates the extremely small probability of transition to the b-Sn phase (under 1 km/s particle velocity) and from our knowledge, there are no experimental techniques able to detect the new phase. Thereby it could be said that the phase transition to b-Sn phase could not occur under such conditions.”
Answering the part of the comment about the maximal depth it is difficult to indicate the depth because only about 10-20 atoms (under different simulation launches) are in the b-Sn phase. The maximal obtained depth for such kind of atoms is 10 A. Amorphization was observed in a layer up to 20 A as it is shown in Fig.7 of the manuscript.
Point 7: Figure 4: The figure caption is wrong.
Answer: We changed the caption
Point 8: Line 175: -Sn -> β-Sn
Answer: We fixed the typo.
Point 9: Figure 6: I recommend to change the unit of color scale on figures 6a and 6b from “bar” to “GPa”. The caption is wrong: … shock wave speed (c,d) and atomic volume (e,f)… -> … atomic volume (c,d) and shockwave velocity (e,f)…
Answer: We changed the scale and fixed the label
Point 10: Figure 7: The scale and label are not clear.
Answer: We redrawn the figure.
Point 11: Line 229: fig. 6c -> fig. 7c
Answer: We fixed the typo
Point 12: Line 255: The -> the (13) Line 282: “…and __ is …”. What?
Answer: We fixed the typo
Point 13: Line 307: particle velociry -> piston speed
Answer: We changed the text
Reviewer 2 Report
The manuscript “Dynamics of ultrafast phase transitions in (001) Si on the shock 2 wavefront” reports a phase transition under ultrafast pressure loading. All data are based on the molecular dynamics simulations for a large sample (about 30x30x30 unit cells or more, for some cases). The authors show atomic volumes, centrosymmetry parameter, and X-ray spectra calculated for two piston speeds. The manuscript is very interesting and well presented. However, the potential reader can find some shortcomings and incomplete analysis of the results. I think that the manuscript can be published in the Journal after major amendments shown below:
- 3, l. 111 and Fig. 2a; How do the authors know that the pressure of phase transition is 11 GPa? They show data for a multiple of 10 GPa. I think that close to the phase transition point they should show more data!
- 3, l. 116 and Fig. 2a; the same question as given above (but for 45 GPa).
- 2a-c; There is given the pressure 0 GPa. Is it simulations done for the vacuum? Or should there be a normal pressure?
- 4, ll. 130-137; the authors should indicate that this part of the text is shown in Fig. 3.
- 4, l. 150; the inserted figures shown in Figs. 4a-b they are not showing “the ratio between shock and particle velocity” but the dependence between these two quantities;
- 4, 5, 6; there is a comma instead of a point as the separator of integer form decimal parts.
- 4 caption is the same as for Fig. 2! Change it!!!
- 5, ll. 173-174; I think that there should be that “a beginning of the transformation to the beta-Sn phase after 200 fs is observed” (?).
- 5, ll. 181-182; there is “(…) the lattice could be destroyed because atoms “forget” their initial position (…)”. I think that this sentence is out of physical meaning. It must be rebuilt! The part of the sentence “atoms “forget” their initial position” sounds strange! I think that the authors should write that the shifts of the atoms from their initial positions will be so huge/large, that the structure will be broken.
- 5, ll. 166-188; the cases for 3 km/s and 1 km/s should be compared within the text.
- 5; the data shown in Figs. 5a, c, e, and 5 b, d, f should be compared within the text and described. It is useless to show any figures without analysing them!
- 5 caption; the shortcut “CSP” is given there but the meaning is missing (on p. 9 this shortcut is explained. It is too late!).
- 5, ll. 197-198; there is “The areas marked as red and orange in Fig 5c, d (…)”, I think that there should be “Fig. 6”;
- 7; the initial temperature should be given there (the temperature before the shock wave starts to propagate). And the melting temperature should be given.
- 8, l. 236; the authors wrote that the simulation was performed on a 30x30x30 unit cell and in Fig. 5 caption there is a 30x30x50 unit cell.
- 8, l. 251; the authors wrote that they equilibrated the system at 305 K. why is the temperature lower than 300 K in Fig. 7a? (and even reaches 0 K) Why does the system reach a temperature of about 250 K in Figs. 7b and 7c?
- 9, Eqs. (2) – (5); in Eqs. (2) and (4) there is “Lp” and in l. 285 “L with subscript p”. In Eq. (2) it should be F(k). There is no K (capital K letter) in equations cited (compare text in l. 281). L. 282 – the letter "theta" is missing.
- 10, ll. 300-308; the conclusions section should be longer.
- Some misprints/errors: (i) 2, l. (l.=line) 61: the authors use a shortcut but at the first time they should give a full name of the ‘X-ray Diffraction method’; (ii) p.2, lines (ll.=lines) 63-64; there is “solid-to-solid” and in l.58 “solid-solid”. It should be unified!; (iii) p.2, l. 73; the author should not use the full name and the shortcut, i.e. “molecular dynamics (MD)” if they give the shortcut in l. 63; (iv) p. 3, ll. 121-122; The beginning of the sentence, i.e. “As the intermediate conclusion of this paragraph, we can propose that (…)” is unnecessary; (v) p. 4, l. 133; there is “ 16A3” and it should be “16 cubic A” (3 as a power); (vi) p. 5, l. 175; the “beta” is missing before “ -Sn”; (vii) p. 5, l. 195; the authors should indicate that they describe data given in Figs. 6; (viii) p. 8, l. 223 there is “4 mk/s piston’s” and there should be “4 km/s piston” (two errors).
Author Response
First of all, we want to thank the Reviewer for their work. All changes in the manuscript are presented in the red-line version of the manuscript. The answers to the issues are given below.
Point 1 and 2: 3, l. 111 and Fig. 2a; How do the authors know that the pressure of phase transition is 11 GPa? They show data for a multiple of 10 GPa. I think that close to the phase transition point they should show more data!
3, l. 116 and Fig. 2a; the same question as given above (but for 45 GPa).
Answer: We determined the phase transitions as a jump in atomic volume, centrosymmetric parameter, and XRD spectra. The simulations were performed with a 0.5 GPa step in the vicinity of the presumptive transition pressure, determined from simulation with a 5 GPa step. We added the additional graph for the phase transition at 11 GPa, where the jump is demonstrated. A similar picture is obtained for other phase transitions. We decided not to add graphs in the vicinity of another phase transitions because it significantly obstructs the figure.
Point 3: 2a-c; There is given the pressure 0 GPa. Is it simulations done for the vacuum? Or should there be a normal pressure?
Answer: The simulation was performed for normal pressure We fixed the label.
Point 4: 4, ll. 130-137; the authors should indicate that this part of the text is shown in Fig. 3.
Answer: We added the reference to Fig.3 in this text fragment.
Point 5: 4, l. 150; the inserted figures shown in Figs. 4a-b they are not showing “the ratio between shock and particle velocity” but the dependence between these two quantities;
Answer: We fixed this sentence.
Point 6: 4, 5, 6; there is a comma instead of a point as the separator of integer form decimal parts.
Answer: We have redrawn these figures
Point 7: 4 caption is the same as for Fig. 2! Change it!!!
Answer: We changed the label.
Point 8: 5, ll. 173-174; I think that there should be that “a beginning of the transformation to the beta-Sn phase after 200 fs is observed” (?).
Answer: We change this text fragment to: “A beginning of the transformation to the beta-Sn phase after 200 fs is observed. The change in centrosymmetry parameter, atomic volume and XRD spectrum associated with the transition to the b-Sn phase is obtained during propagation of the shock wave.”
Point 9: 5, ll. 181-182; there is “(…) the lattice could be destroyed because atoms “forget” their initial position (…)”. I think that this sentence is out of physical meaning. It must be rebuilt! The part of the sentence “atoms “forget” their initial position” sounds strange! I think that the authors should write that the shifts of the atoms from their initial positions will be so huge/large, that the structure will be broken.
Answer: We change the sentence to: “If the amplitude of the shock wave would be higher, the shifts of the atoms from their initial positions will be so huge, that the structure will be broken, the lattice could be destroyed, and the Si will become amorphous”
Point 10: 5, ll. 166-188; the cases for 3 km/s and 1 km/s should be compared within the text.
Answer: We compared 3km/s and 1 km/s case and added the following text fragment:
Under 1 km/s piston velocity, no significant jumps in the atomic volume were detected. The analysis shows that only a small amount (less than 10-6) of atoms is in the b-Sn phase, in the histograms, it manifests itself as a broadening a peak at 20 A3 to lower volumes (see Fig.3a), in contradistinction to 3 km/s piston velocity, when a new peak at 16 A3 is formed. Such a tiny amount of new phase in our opinion is not enough to assert the possibility of a phase transition a-diamond => b-Sn under 1 km/s piston velocity.
Point 11: 5; the data shown in Figs. 5a, c, e, and 5 b, d, f should be compared within the text and described. It is useless to show any figures without analysing them!
Answer: We added discussion to the text of the MS, for example, see lines 191,192, 199, etc.
Point 12: 5 caption; the shortcut “CSP” is given there but the meaning is missing (on p. 9 this shortcut is explained. It is too late!).
Answer: we change the first mention of the acronym to line 79.
Point 13: 5, ll. 197-198; there is “The areas marked as red and orange in Fig 5c, d (…)”, I think that there should be “Fig. 6”;
Answer: We fixed the typo.
Point 14: 7; the initial temperature should be given there (the temperature before the shock wave starts to propagate). And the melting temperature should be given.
Answer: We added the value of the initial temperature and melting point to the caption of the Fig.7. and to the text
Point 15: 8, l. 236; the authors wrote that the simulation was performed on a 30x30x30 unit cell and in Fig. 5 caption there is a 30x30x50 unit cell.
Answer: In the stationary case the simulation was performed for 30x30x30 cell and 30x30x390 cell in dynamical case (as it mentioned in Methods). The histograms of atomic volume distribution etc. were obtained for a layer containing 50 atoms in z-direction. To avoid further misunderstanding we added the following text: “30×30×50-unit cells i.e., a layer containing 50 atom layers”
Point 16: 8, l. 251; the authors wrote that they equilibrated the system at 305 K. why is the temperature lower than 300 K in Fig. 7a? (and even reaches 0 K) Why does the system reach a temperature of about 250 K in Figs. 7b and 7c?
Answer: The color map was picked up in such a manner to be the most contrasting. And the lowest temperature is higher than 250. It is about 280 K. Under piston impact, few atoms could leave the sample surface, and their temperature could be lower than the initial one.
Point 17: 9, Eqs. (2) – (5); in Eqs. (2) and (4) there is “Lp” and in l. 285 “L with subscript p”. In Eq. (2) it should be F(k). There is no K (capital K letter) in equations cited (compare text in l. 281). L. 282 – the letter "theta" is missing.
Answer: We fixed this text fragment.
Point 18: 10, ll. 300-308; the conclusions section should be longer.
Answer: We slightly increase the conclusion; however, we believe that conclusion represents the main results of the publication.
Point 19: Some misprints/errors: (i) 2, l. (l.=line) 61: the authors use a shortcut but at the first time they should give a full name of the ‘X-ray Diffraction method’; (ii) p.2, lines (ll.=lines) 63-64; there is “solid-to-solid” and in l.58 “solid-solid”. It should be unified!; (iii) p.2, l. 73; the author should not use the full name and the shortcut, i.e. “molecular dynamics (MD)” if they give the shortcut in l. 63; (iv) p. 3, ll. 121-122; The beginning of the sentence, i.e. “As the intermediate conclusion of this paragraph, we can propose that (…)” is unnecessary; (v) p. 4, l. 133; there is “ 16A3” and it should be “16 cubic A” (3 as a power); (vi) p. 5, l. 175; the “beta” is missing before “ -Sn”; (vii) p. 5, l. 195; the authors should indicate that they describe data given in Figs. 6; (viii) p. 8, l. 223 there is “4 mk/s piston’s” and there should be “4 km/s piston” (two errors).
Answer: We fixed these typos.
Round 2
Reviewer 2 Report
The authors answered all my questions/doubts/suggestions, and amended figures pointed out. The manuscript should be published in present form (without any amendments). I think that authors can mark (in the figure captions – 2 and 5) that the curves presented are vertically shifted to get visibility of the data presented.